# Current Concepts in Viscosupplementation: New Classification System and Emerging Frontiers

**DOI:** 10.3390/bioengineering12101050

**Published:** 2025-09-29

**Authors:** Gustavo Constantino de Campos, Alberto Cliquet

**Affiliations:** Departamento de Ortopedia, Reumatologia e Traumatologia, Universidade Estadual de Campinas, Campinas 13083-887, Brazil; cliquet@unicamp.br

**Keywords:** knee osteoarthritis, viscosupplementation, hyaluronic acid, hyaluran receptors, injections, intra-articular, classification

## Abstract

Viscosupplementation with intra-articular hyaluronic acid (HA) is a key therapeutic option for osteoarthritis (OA), yet the field is hampered by clinical controversies and an outdated classification of available products. This comprehensive review critically analyzes the current landscape, moving from a mechanical to a biological paradigm of HA’s mechanism of action. We argue that the traditional HA product classification based solely on molecular weight is insufficient, as it conflates chemically distinct products. Therefore, we propose a new, two-tiered classification framework: the primary distinction is based on chemical structure, separating linear (non-modified) HA from cross-linked (chemically modified) HA. Linear HA is then sub-classified by molecular weight (Low, Intermediate, and High), while cross-linked HA is defined as a separate category of hydrogels with a ultra-high effective molecular weight. Within this clearer framework, we analyze the central controversy between formulations, highlighting the pivotal emergence of high-concentration, high-molecular-weight (>2 million Dalton) linear HA. These formulations not only challenge the durability rationale for cross-linking by providing year-long efficacy but also possess a superior biological profile for chondroprotection, preserving chondrocyte viability and function. Furthermore, we explore the expanding frontier of combination therapies, where linear HA serves as the ideal physiological scaffold for agents like corticosteroids, PRP and other injectable orthobiologics such as bone marrow aspirate and stromal vascular fraction.

## 1. Introduction

Osteoarthritis (OA) is the most prevalent degenerative joint disease and a primary cause of disability worldwide, particularly in aging populations [1]. Its societal and economic burden is immense, with costs related to treatment and lost productivity estimated to be as high as 2.5% of the gross domestic product in developed nations [2]. As a progressive condition, OA is characterized by the degradation of articular cartilage, pathological remodeling of subchondral bone, and synovial inflammation, leading to significant pain, loss of function, and a reduced quality of life [3]. The knee is one of the most commonly affected joints, and the global prevalence of knee OA is rising, driven not only by an aging population but also by an increasing incidence of risk factors such as obesity, previous joint injuries, and genetic predispositions [1,2,4].

A major challenge in the clinical management of knee OA is that a diagnosis is often made only after significant structural damage has occurred. The early stages of cartilage degradation are frequently asymptomatic, and a poor correlation exists between radiographic findings and the patient’s subjective experience of pain [4]. In an osteoarthritic joint, the synovial fluid undergoes critical rheological changes, including a reduction in the concentration and molecular weight of its main component, hyaluronic acid (HA) [5]. This degradation compromises the fluid’s ability to lubricate the joint and absorb mechanical shock, thereby contributing to the perpetuation of the pain and degeneration cycle [6].

Viscosupplementation aims to counteract these changes through the intra-articular injection of exogenous HA [7]. Despite its long-standing use, the field suffers from a lack of consensus, driven by conflicting clinical data and a confusing product landscape. This divergence is reflected in international clinical practice guidelines, which vary significantly in their recommendations [8]. Clinical practice guidelines from renowned societies, such as the American Academy of Orthopaedic Surgeons (AAOS), consistently recommend against its use for knee OA, citing a lack of clinically significant efficacy [9]. Meanwhile, others, like the Osteoarthritis Research Society International (OARSI), maintain it as a valid option for specific patient subgroups [10]. A major contributor to this confusion is an obsolete classification system that fails to distinguish between fundamentally different types of HA products.

This comprehensive review aims to dissect the current concepts in viscosupplementation, not only by analyzing the evidence but also by providing a clearer lens through which to view it. This article will first explore the dual mechanism of action of HA, which involves not only a restoration of the synovial fluid’s viscosity and elasticity, but presents direct interaction with cellular receptors like CD44 to modulate inflammation, stimulate endogenous HA production, and exert direct chondroprotective effects [11,12,13]. Furthermore, we explore the expanding frontier of combination therapies, where linear HA serves as the ideal physiological scaffold for agents like platelet-rich plasma (PRP) and corticosteroids. We will then challenge the traditional division of viscosupplementation products primarily into native vs. cross-linked and low-molecular-weight vs. high-molecular-weight formulations [14] and propose a new, more clinically and biologically relevant framework.

## 2. Viscosupplementation’s Dual Mechanism of Action: Biology and Rheology

The original premise of viscosupplementation was simple and elegant: to replace the “defective” HA in the arthritic joint to restore the synovial fluid’s viscosity and elasticity. This joint “lubrication” would reduce friction between cartilage surfaces, absorb shock during load-bearing, and consequently alleviate pain. While these mechanical effects are undeniable, they fail to fully explain the observed clinical benefits, especially the duration of pain relief, which often far exceeds the short intra-articular residence time of the injected HA (from days to a few weeks) [15].

The therapeutic effect of viscosupplementation rests on two distinct pillars: biological activity and rheological function. The most important rheological property is viscosity, a fluid’s resistance to flow [16]. In a healthy joint, synovial fluid is exquisitely viscoelastic: viscous for low-friction lubrication during slow movement and elastic for shock absorption during high-impact activities [17]. In OA, the loss of HA impairs both functions [6]. While restoring these mechanical properties is an immediate goal, the long-term benefits of HA are primarily biological. Modern research has revealed a predominantly biological mechanism of action. Exogenous HA interacts directly with the joint’s biology through multiple pathways:

### 2.1. Interaction with Cellular Receptors

HA binds to receptors on the surface of synoviocytes, chondrocytes, and immune cells, such as CD44 and the Receptor for Hyaluronan-Mediated Motility (RHAMM) [18]. This interaction with receptors like CD44 and RHAMM is highly dependent on the molecular weight of the HA chain [19]. Binding to these receptors initiates a cascade of intracellular signals that directly influence gene expression [18]. Specifically, this can upregulate the synthesis of essential cartilage matrix components, such as type II collagen and aggrecan, while simultaneously downregulating the genes responsible for inflammatory mediators and matrix-degrading enzymes [18]. This signaling also plays a role in cellular senescence, a state of irreversible cell cycle arrest implicated in OA pathogenesis. By interacting with senescent cells, HA may act as a “senomorphic” agent, helping to restore a healthier joint environment [13]. Therefore, the HA molecule is not a passive substance but an active signaling molecule that helps steer the joint environment away from a catabolic (breakdown) state and toward an anabolic (build-up) state.

### 2.2. Anti-Inflammatory Effect

Through these signaling pathways, HA exerts a potent anti-inflammatory effect. It has been shown to inhibit the production and activity of key pro-inflammatory cytokines like Interleukin-1β (IL-1β), Tumor Necrosis Factor-α (TNF-α), chemokines, and matrix metalloproteinases (MMPs), which are enzymes responsible for cartilage degradation [12,20]. This includes the inhibition of MMP-1, MMP-3, and MMP-13, which are critical drivers of collagen and proteoglycan breakdown in OA cartilage [21]. Furthermore, exogenous HA can modulate immune cell behavior within the joint, reducing the infiltration of macrophages and polarizing them towards an anti-inflammatory M2 phenotype, away from the pro-inflammatory M1 state [20].

Beyond inflammation, HA also mitigates oxidative stress, another key driver of OA pathology. It directly scavenges reactive oxygen species (ROS) and reduces the production of nitric oxide (NO), protecting chondrocytes from oxidative damage and apoptosis [21]. By suppressing these inflammatory pathways, HA helps to break the cycle of synovitis and cartilage degradation that characterizes osteoarthritis. Moreover, evidence suggests that the immunomodulatory capacity of hyaluronic acid is dependent on its molecular weight and concentration. Indeed, one study demonstrated that a high-concentration, linear formulation with a molecular weight exceeding 1.2 million Daltons (MiDa) was the most effective at reducing inflammation [11].

### 2.3. Analgesic Effect

Hyaluronic acid (HA) alleviates osteoarthritic pain through multiple molecular pathways. It can reduce pain by decreasing the production of algesic substances like Prostaglandin E2 and Bradykinin, and also by physically masking and inhibiting the sensitization of nociceptors present in the synovium [11,12]. It also directly engages with nociceptive receptors such as the transient receptor potential vanilloid 1 (TRPV1) and acid-sensing ion channels (ASICs), which are key mediators of pain sensation. By creating a protective coating over these nerve endings, HA shields them from the acidic and inflammatory environment of the osteoarthritic joint [21]. Furthermore, HA has been observed to decrease the release of substance P, a critical neuropeptide in the transmission of pain signals and the inflammatory cascade [21].

### 2.4. Endogenous Stimulation (Chondroprotection)

One of the most clinically significant biological effects is the ability of exogenous HA to stimulate type B synoviocytes to increase their own production of high-molecular-weight HA. This is a crucial finding, as it suggests that viscosupplementation is not merely a replacement therapy but a biological ‘trigger’ that helps the joint restore its own natural homeostasis and lubrication over a prolonged period. This process, often termed ‘endogenous stimulation,’ helps explain why the clinical benefits of HA can last for months, far exceeding the physical residence time of the injected product. This effect is most pronounced with linear HA formulations, which, being identical to the native molecule, are more readily recognized by the cellular machinery responsible for this feedback loop [19,22].

## 3. Revisiting the Classification of Hyaluronic Acid Formulations

A significant barrier to clarity in the field has been its antiquated classification system. Traditionally, products were categorized by molecular weight (MW): Low MW (<1 MiDa), Intermediate MW (1–2 MiDa), and High MW (>6 MiDa) [15]. This system is fundamentally flawed because it places chemically modified hydrogels and natural polymers in the same category, creating confusion.

Furthermore, a simple dichotomy between native (linear) and cross-linked formulations is also an oversimplification [14], since the linear HA category is not homogenous. Molecular weight is a critical determinant that separates products with poor clinical performance from those with excellent outcomes.

To solve this, we propose a more logical, two-tiered classification framework. The primary distinction is based on the most fundamental product characteristic—its chemical structure—while the secondary tier considers the crucial role of molecular weight.

### Chemical Structure (The Primary Distinction)

The first and most important classification is between products based on their intrinsic chemical makeup.

**(A)** **Linear HA (Non-modified):** Consists of non-modified hyaluronic acid chains, identical to those found endogenously [5]. Their defining characteristics are superior biocompatibility and a physiological mechanism of action. They can be further sub-classified by their molecular weight:

Low Molecular Weight (LMW) Linear HA: <1 MiDa;

Intermediate Molecular Weight (IMW) Linear HA: 1–2 MiDa;

High Molecular Weight (HMW) Linear HA: >2 MiDa;

**(B)** **Cross-linked HA (Chemically Modified Hydrogels):** Consists of HA chains that have been chemically bonded (e.g., using BDDE) to form a synthetic hydrogel. This modification creates products with a **Ultra-High effective Molecular Weight (UHMW)**, typically cited as >6 MiDa. This places them in a distinct category defined by their non-physiological structure, altered biocompatibility, and different safety profile.

This proposed framework (Table 1) provides immediate clarity. It properly isolates cross-linked products into their own category and forces a distinction based on the most clinically relevant question: is the molecule physiological or chemically modified?

## 4. The Evolving Controversy (Linear vs. Cross-Linked Products) in Light of the New Classification

Classically, the primary limitation of natural or linear HA was considered to be its rapid degradation by hyaluronidase and free radicals within the joint, with a half-life of only a few hours [23]. To overcome this, the industry developed second-generation products using cross-linking technology. In this process, linear HA molecules are chemically bonded by agents such as BDDE (1,4-butanediol diglycidyl ether), forming a more robust hydrogel network with a higher molecular weight. The most well-known examples are Hylan G-F 20 and NASHA Technologies [14]. The rationale is straightforward: by creating a larger molecule that is more resistant to degradation, the intra-articular residence time is increased from days to weeks. Theoretically, this would prolong both the mechanical viscoelastic “cushioning” effect and the time available for the molecule to exert its biological effects [24]. However, despite the appealing logic, the chemical modification of HA is not without potential disadvantages, which form the basis of the argument in favor of more physiological, linear products.

### 4.1. Clinical Effectiveness

The controversy surrounding the clinical effectiveness of viscosupplementation is largely fueled by the heterogeneity of the available evidence. Results from clinical trials and subsequent meta-analyses have been inconsistent, leading to divergent recommendations in clinical practice guidelines [10,25,26]. This variability can be attributed to several factors, including the pooling of data from studies with diverse patient populations, different criteria for osteoarthritis severity (e.g., Kellgren-Lawrence grades I–IV), and varied administration regimens (e.g., single vs. multiple injections). Crucially, many large-scale analyses fail to differentiate between the distinct biochemical properties of the various HA products. This methodological issue has contributed to conclusions from highly influential meta-analyses, such as the 2022 review by Pereira et al. and the analysis informing the American Academy of Orthopaedic Surgeons (AAOS) guidelines, which found that the average treatment effect of HA did not meet the threshold for clinical significance, thus recommending against its routine use [9,26].

Nonetheless, some comparative studies and meta-analyses have suggested a good effectiveness profile, with a slight advantage in the magnitude of pain relief for higher molecular weight products [27,28]. This finding has been traditionally attributed to the superior viscoelastic properties and longer intra-articular residence time of these larger molecules. However, a significant confounding factor in these broad comparisons is the inclusion of low-molecular-weight linear products, which are known to be clinically inferior. There is abundant literature demonstrating that low-molecular-weight linear HA (<1 MiDa) is associated with lower efficacy [27,28], suboptimal interaction with CD44 receptors [19], and even potential pro-inflammatory properties [29], thereby skewing the overall analysis against the entire class of linear products.

When the analysis is refined to focus specifically on high-molecular-weight linear HA, the evidence reveals a consistent and robust clinical efficacy that rivals—and in some cases may surpass—that of cross-linked formulations. For instance, studies on 1% sodium hyaluronate with a molecular weight of 2.4–3.6 MiDa have repeatedly shown significant improvements in pain and function that are sustained for at least six months [30,31]. Furthermore, a recent single-injection, high-concentration (2.5%) linear HA product demonstrated sustained efficacy for up to one year, with a significant reduction in pain and a notable improvement in quality of life, underscoring the long-term clinical potential of non-cross-linked HA [32,33]. This body of evidence strongly suggests that once a therapeutic threshold of molecular weight and concentration is achieved, the superior biocompatibility and biological activity of a linear structure become paramount, offering a powerful and durable clinical effect without the need for chemical modification.

### 4.2. Biocompatibility and Safety

The main argument against cross-linked products is the increased risk of acute local inflammatory reactions. Numerous studies and case reports have documented the occurrence of severe pain, swelling, and effusion (acute synovitis or “pseudosepsis”) following injections of Hylan G-F 20 [34,35,36], and even foreign body reactions following NASHA injection [37]. Recent case reports have provided morphological evidence of hyaluronan-related granulomatous reactions, where injected material is found not just in the synovium but also in the infrapatellar fat pad and even within the subchondral bone, acting as a foreign body and provoking a chronic inflammatory response [38], complications that are much rarer with linear products. The hypothesis is that the chemically modified molecule and hydrogel particulates can be recognized as a foreign body by the host’s immune system, triggering a robust inflammatory response. Being identical to endogenous HA, linear products possess an inherently superior biocompatibility and safety profile.

### 4.3. Physiological Biological Activity: Beyond Symptom Relief

While the debate between linear and cross-linked HA has often centered on durability and safety, a critical dimension that favors linear HA is its direct chondroprotective potential. Chondroprotection, in this context, refers to the ability to slow the progressive degradation of articular cartilage and potentially promote repair mechanisms.

Modern evidence increasingly points to the role of HMW linear HA in preserving chondrocyte viability and function. In vitro studies have demonstrated that linear HA can protect chondrocytes from apoptosis (programmed cell death) induced by inflammatory or oxidative stress. It achieves this by reducing the expression of key catabolic enzymes, particularly the aggrecanases (e.g., ADAMTS-5) and matrix metalloproteinases (MMPs) that are responsible for cleaving the essential components of the cartilage matrix [39].

Furthermore, the superior biological interaction of linear HA with cellular receptors is pivotal for this chondroprotective effect. By optimally engaging with receptors like CD44, linear HA can more effectively stimulate chondrocytes to synthesize new matrix proteins, such as proteoglycans and type II collagen. The cross-linking process, by altering the molecule’s three-dimensional structure, may interfere with its ability to optimally interact with cellular receptors like CD44 [27]. Research suggests that HA in the intermediate molecular weight range (0.5 to 3.0 MiDa), characteristic of many linear products, possesses the maximal biological activity for stimulating endogenous HA production and promoting chondroprotection [19]. A giant, cross-linked product might be less “biologically recognized,” acting more as a passive implant than as a bioactive agent.

### 4.4. Natural Degradation

Linear products are broken down by the body’s natural enzymatic pathways. In contrast, the degradation of cross-linked gels is less physiological and could, theoretically, release residuals of the cross-linking agents.

Moreover, the historical argument for Cross-linked HA (superior durability) is being dismantled by advancements in the HMW Linear HA category. The development of HMW Linear HA products with high concentrations (up to 2.5%) has been a game-changer. The increased concentration enhances the entanglement of the polymer chains, significantly boosting the solution’s viscosity and resistance to degradation without any chemical modification. This innovation allows HMW Linear HA to achieve a much longer intra-articular residence time. Objective evidence from a vibroarthrography study confirmed that a single injection of a 2.5% sodium hyaluronate solution containing 120 mg of linear HA led to an immediate and sustained improvement in joint motion quality, directly correlating with clinical improvements in pain and function [40]. Furthermore, post-market clinical follow-up studies confirm that a single injection can provide significant and clinically relevant improvements in pain and function that are maintained for a full 12-month period and beyond [32,33]. Therefore, a linear product can achieve the single-injection, long-duration profile that was once the exclusive domain of cross-linked formulations. This finding effectively makes the primary argument for using Cross-linked HA obsolete.

## 5. Emerging Frontiers: Combination Therapies with Hyaluronic Acid

While the choice of HA formulation is a critical primary decision, the clinical frontier is expanding to include combination therapies designed to harness synergistic effects. The role of HA as a biologically active scaffold makes it an ideal partner for other intra-articular treatments, most notably corticosteroids, PRP and other orthobiologics.

### 5.1. Co-Formulations with Corticosteroids

The rationale for combining HA with a corticosteroid is to achieve both rapid and sustained pain relief. Corticosteroids are potent anti-inflammatory agents that can quickly control acute synovitis and provide powerful, short-term analgesia. However, their benefits are often transient, and concerns exist regarding potential chondrotoxicity with repeated use [41].

By combining the two, a single injection can provide the immediate anti-inflammatory ‘burst’ from the corticosteroid to manage an acute flare-up, while the HA provides its long-term benefits: sustained lubrication, biological modulation, and chondroprotection. This strategy allows the HA to begin working in a less inflamed environment, potentially enhancing its efficacy. It has been shown that the addition of triamcinolone improves short-term results of viscosupplementation without altering its adverse effects [42].

### 5.2. Association with Platelet-Rich Plasma (PRP) and Other Orthobiologics

The role of hyaluronic acid as a biological scaffold extends to combinations with a range of orthobiologics, which represent the frontier of regenerative medicine for osteoarthritis. These therapies aim not just to manage symptoms but to modulate the intra-articular environment to favor tissue repair over degradation.

Platelet-Rich Plasma (PRP) is an autologous blood concentrate rich in a multitude of growth factors (e.g., TGF-β, PDGF, VEGF) and signaling molecules that are pivotal for tissue healing and regeneration [43]. The therapeutic synergy between PRP and HA is compelling: PRP provides the bioactive “signals” (growth factors) for tissue repair, while HA provides the ideal “soil” (a biocompatible and bioactive scaffold) [44]. At a molecular level, the combination is highly logical. PRP’s growth factors work to counteract the inflammatory and catabolic environment of the OA joint by inhibiting inflammatory signaling pathways (like NF-κB) and reducing the production of pro-inflammatory cytokines [45]. It also promotes the proliferation and differentiation of chondrocytes [46]. When combined, HA creates a viscous, protective environment that can prolong the residence time and bioavailability of these growth factors, creating a more favorable and sustained milieu for them to exert their regenerative effects on chondrocytes and synoviocytes [47].

The clinical rationale for this combination is to leverage the strengths of both agents. While HA provides excellent viscosupplementation and lubrication, PRP offers a stronger, more direct anti-inflammatory and pro-anabolic effect [48]. Multiple studies and meta-analyses have shown that while both treatments are effective, PRP may offer more sustained pain relief and functional improvement over time compared to HA alone [49]. A recent network meta-analysis of 75 randomized controlled trials confirmed that PRP was the most effective injectable for improving function at 12 months [50]. Furthermore, a systematic review focusing specifically on the combination therapy concluded that the intra-articular injection of PRP combined with HA is more effective for pain relief and functional improvement than HA or PRP alone, with benefits observed at 3, 6, and 12-month follow-ups [47]. This suggests a true synergistic effect rather than just an additive one. A novel approach involving the administration of PRP one week before HA has also shown promising results, theorizing that PRP first modulates the inflammatory environment, making the joint more receptive to the subsequent benefits of HA [51].

Beyond PRP, this synergistic concept is being extended to cell-based therapies, including Mesenchymal Stem/Stromal Cells (MSCs). MSCs possess potent immunomodulatory and regenerative capabilities, acting through paracrine signaling to release a host of anti-inflammatory and trophic factors that can protect cartilage and promote a healthy joint environment [52]. In these applications, HA serves a critical function not just as a delivery vehicle, but as a pro-survival scaffold. The viscous, biocompatible hydrogel created by linear HA protects the implanted cells from the harsh mechanical and inflammatory environment of the osteoarthritic joint. This protection is crucial for enhancing their viability, engraftment, and paracrine signaling capabilities [52]. Combining MSCs with a supportive scaffold like HA has been shown to improve their therapeutic efficacy compared to injecting the cells alone [52].

Another emerging area is the use of minimally manipulated adipose tissue products, such as Adipose-Derived Stromal Vascular Fraction (AD-SVF). This tissue is a rich source of MSCs, pericytes, and other regenerative cells [53].

AD-SVF injections provide a complex cocktail of cells and growth factors that work together to reduce inflammation, inhibit cartilage degradation, and stimulate tissue repair [53]. Clinical studies comparing adipose-derived products to PRP have suggested that both are effective, with some evidence indicating that the adipose-derived treatments may provide more durable long-term benefits in pain and function [54]. The co-administration with HA is a logical next step, applying the same principle of using HA as a protective and synergistic scaffold to enhance the viability and function of the transplanted regenerative cells.

In all of these combination strategies, the choice of HA is critical. A biocompatible, non-modified linear HA is the most logical partner, serving as an ideal physiological scaffold that enhances the action of the biologic agent without introducing the confounding variables and altered biological response associated with chemical cross-linking. This positions linear HA as a foundational element in the future of combination regenerative therapies for osteoarthritis.

## 6. Conclusions

The field of viscosupplementation requires a clearer, more rational approach to both product classification and clinical decision-making. The traditional weight-based classification system is inadequate. We have proposed a new, two-tiered framework that: firstly distinguishes products based on their fundamental chemical structure—linear versus cross-linked—a distinction of paramount biological and clinical importance; secondly distinguishes the linear products in 3 categories of molecular weight.

According to this new classification, low-molecular-weight linear products may be considered suboptimal, as they are characterized by reduced efficacy and a pro-inflammatory nature. Conversely, both high-molecular-weight linear and cross-linked products exhibit a more favorable durability and efficacy profile. Of these, the linear products present a distinct advantage, as they are not chemically modified.

Future research should focus on head-to-head trials comparing HMW linear HA against cross-linked HA, as well as pharmacoeconomic analyses, to further solidify its definitive role in the management of osteoarthritis.

## Figures and Tables

**Table 1 bioengineering-12-01050-t001:** Classification of hyaluronic acid products.

Category	Acronym	Chemical Structure	Molecular Weight
Low Molecular Weight Linear HA	**LMW**		<1 MiDa
Intermediate Molecular Weight Linear HA	**IMW**	Linear	1–2 MiDa
High Molecular Weight Linear HA	**HMW**		>2 MiDa
Ultra High Molecular Weight Cross-linked HA	**UHMW**	Cross-linked	>6 MiDa

## Data Availability

No new data were created or analyzed in this study.

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
