# Peer review of "Current Concepts in Viscosupplementation: New Classification System and Emerging Frontiers"

_bioengineering, 2025, doi:10.3390/bioengineering12101050_

Round 1

Reviewer 1 Report

Comments and Suggestions for Authors

The manuscript proposes a classification system for hyaluronic acid (HA) based on its chemical structure (linear vs. cross-linked). I have comments listed below:

  1. While the distinction between linear and cross-linked HA is clinically relevant, the manuscript does not introduce a truly novel perspective on HA classification. The linear vs. cross-linked divide has already been explored in multiple publications, and the existing literature has extensively addressed the immunologic reactions associated with cross-linked HA. Therefore, the manuscript would benefit from a clearer indication of how this classification advances current understanding or offers new insights into HA's clinical applications.
  2. The manuscript references some earlier works on HA classification but overlooks recent comprehensive reviews that could significantly enhance its credibility. Specifically, the articles by Salih (2024) and Iaconisi (2023) should be included, as they offer more up-to-date insights into HA formulations and their clinical application in viscosupplementation.
  3. Please provide a more detailed explanation of how different HA formulations, specifically linear versus cross-linked, interact with cellular receptors such as CD44 and RHAMM, and the downstream effects.
  4. The manuscript should present more research evidence to demonstrate the advantages of high-molecular-weight (HMW) linear HA over cross-linked HA products, particularly in terms of clinical outcomes such as pain relief, joint function, and long-term efficacy.
  5. Please provide information on how the new classification framework could potentially guide clinical decision-making for the clinical application of HA treatment or improve patient outcomes.

Author Response

Dear reviewer,

Thank you so much for your comments.

Here is a point-by-point response to them:

  1. While the distinction between linear and cross-linked HA is clinically relevant, the manuscript does not introduce a truly novel perspective on HA classification. The linear vs. cross-linked divide has already been explored in multiple publications, and the existing literature has extensively addressed the immunologic reactions associated with cross-linked HA. Therefore, the manuscript would benefit from a clearer indication of how this classification advances current understanding or offers new insights into HA's clinical applications.

RESPONSE: We modified introduction and section 3 to a clear indication

  1. The manuscript references some earlier works on HA classification but overlooks recent comprehensive reviews that could significantly enhance its credibility. Specifically, the articles by Salih (2024) and Iaconisi (2023) should be included, as they offer more up-to-date insights into HA formulations and their clinical application in viscosupplementation.

RESPONSE: We’ve updated the references including Iacosini (2023) and other relevant articles

  1. Please provide a more detailed explanation of how different HA formulations, specifically linear versus cross-linked, interact with cellular receptors such as CD44 and RHAMM, and the downstream effects.

RESPONSE: The cross-linking process, by altering the molecule's three-dimensional structure, may interfere with its ability to optimally interact with cellular receptors like CD44 [23]. Research suggests that HA in the intermediate molecular weight range (0.5 to 3.0 Million Daltons), characteristic of many linear products, possesses the maximal biological activity for stimulating endogenous HA production and promoting chondroprotection [16]. A giant, cross-linked product might be less "biologically recognized," acting more as a passive implant than as a bioactive agent. (lines 260-266)

  1. The manuscript should present more research evidence to demonstrate the advantages of high-molecular-weight (HMW) linear HA over cross-linked HA products, particularly in terms of clinical outcomes such as pain relief, joint function, and long-term efficacy.

RESPONSE: we added a section to address your comment – section 4.1 Clinical Effectiveness (lines 197-232)

  1. Please provide information on how the new classification framework could potentially guide clinical decision-making for the clinical application of HA treatment or improve patient outcomes.

RESPONSE: According to this new classification, low-molecular-weight linear products may be considered suboptimal, as they are characterized by reduced efficacy and a proinflammatory nature. Conversely, both high-molecular-weight linear and cross-linked products exhibit a more favorable durability and efficacy profile. Of these, the linear products present a distinct advantage, as they are not chemically modified. (lines 372-376)

Reviewer 2 Report

Comments and Suggestions for Authors

The article presents a current overview of concepts related to viscosupplementation with hyaluronic acid (HA) in the treatment of osteoarthritis of the knee. The authors propose a new classification of HA preparations, based primarily on their chemical structure (linear vs. cross-linked HA) and then on their molecular weight. They also point to the growing importance of combination therapies (e.g., HA + PRP, HA + glucocorticosteroids).

The paper is well written, logically structured, and based on extensive literature. It stands out for its attempt to organize the complex issue of HA preparation classification and for drawing attention to the biological potential and safety of individual forms. It is a valuable source of knowledge for both clinicians and researchers involved in the treatment of osteoarthritis. Before further processing, the article will require minor corrections and additions. Detailed comments are provided below.

Minor comments:

Expanding the introduction with a more comprehensive discussion of osteoarthritis would enhance the section by emphasizing both its clinical relevance and its societal burden. Osteoarthritis is a complex, multifactorial disorder shaped by factors such as occupational stress, sports activity, prior injuries, obesity, and sex-related differences. Providing a concise overview of these risk factors, together with the clinical context of knee osteoarthritis—particularly the significance of early diagnosis and the limitations of subjective clinical evaluation—would establish a strong foundation for the study. To strengthen this part, the following references are recommended: https://doi.org/10.3390/app15126896; DOI: 10.3390/healthcare12161648 ;

The authors clearly favor linear HMW preparations, describing them as “the most logical therapeutic choice.” There is a lack of a more balanced discussion of the limitations and clinical situations in which cross-linked preparations may have an advantage. In my opinion, the discussion should be supplemented with a critical review of the clinical evidence and a more balanced summary should be presented.

The article is a narrative review, but it does not describe the methodology used to search for and select the literature. This undermines the transparency and reproducibility of the analysis. The methods section should be supplemented with a brief description of the search strategy (databases, years, keywords).

The authors mention the controversy surrounding the effectiveness of HA, but there is no detailed analysis of the differences in study results (e.g., diverse patient populations, OA severity criteria, different administration regimens). The discussion section should be expanded to discuss the sources of heterogeneity and their impact on the interpretation of the results.

The paper does not address the economic issues that are important in the choice of OA therapy. Please at least briefly refer to the available pharmacoeconomic analyses or indicate the need to conduct them.

Although the article discusses biological and clinical aspects, there is a lack of information about the availability of various HA preparations on the market, registration requirements, or differences between countries. Please supplement the work with a practical element that will facilitate the application of the conclusion in everyday medical practice.

In section 4, there are two subsections marked as 4.1. Please correct the numbering to maintain a logical and consistent structure of the paper.

There are typos and minor linguistic errors in the text (e.g., “Futhermore” → “Furthermore,” “foreing” → “foreign,” ‘Classicaly’ → “Classically,” “It is been shown” → “It has been shown”). Please proofread the entire manuscript.

Repetitive and overly categorical phrases (“the conclusion is evident,” “most advanced and logically superior”) are noticeable. I suggest softening the tone towards a more balanced scientific style.

Please standardize the spelling: “intra-articular” vs. “intraarticular.”

Abbreviations and definitions should be used consistently – for example, UHMW (Ultra-High Molecular Weight) should be defined uniformly and used throughout the text.

Table 1 requires correct formatting (header, column layout, consistent use of units).

For greater clarity, I recommend adding diagrams or figures (e.g., comparison of linear and cross-linked preparations, HA mechanism of action).

There are inconsistencies in the formatting of the reference list: different abbreviations for journals, no spaces after commas, inconsistent presentation of DOI numbers (some works include DOI, others do not). Please standardize in accordance with the journal's style.

In the “Conflicts of Interest” section, the phrase “The author declares...” appears, while the article has two authors. Please correct it to “The authors declare...”.

Please standardize the notation of units and numerical values – in some parts of the text, “MDa” is used, while in other places the verbal notation (“Million Dalton”) is used. It is recommended to use a uniform abbreviated form.

Please ensure that all abbreviations are expanded when first used in the text and used consistently.

After these modifications, the paper will be able to serve as a solid reference point for future research and clinical practice.

Comments on the Quality of English Language

There are typos and minor linguistic errors in the text (e.g., “Futhermore” → “Furthermore,” “foreing” → “foreign,” ‘Classicaly’ → “Classically,” “It is been shown” → “It has been shown”). Please proofread the entire manuscript.

Author Response

Dear reviewer,

Thank you so much for your comments.

Here is a point-by-point response to them:

  1. Expanding the introduction with a more comprehensive discussion of osteoarthritis would enhance the section by emphasizing both its clinical relevance and its societal burden. Osteoarthritis is a complex, multifactorial disorder shaped by factors such as occupational stress, sports activity, prior injuries, obesity, and sex-related differences. Providing a concise overview of these risk factors, together with the clinical context of knee osteoarthritis—particularly the significance of early diagnosis and the limitations of subjective clinical evaluation—would establish a strong foundation for the study. To strengthen this part, the following references are recommended: https://doi.org/10.3390/app15126896; DOI: 10.3390/healthcare12161648 ;

RESPONSE:  We’ve expanded the introduction as requested and added those 2 references.

  1. The authors clearly favor linear HMW preparations, describing them as “the most logical therapeutic choice.” There is a lack of a more balanced discussion of the limitations and clinical situations in which cross-linked preparations may have an advantage. In my opinion, the discussion should be supplemented with a critical review of the clinical evidence and a more balanced summary should be presented.

RESPONSE: We agree it was unbalanced and presented a more balanced summary

  1. The article is a narrative review, but it does not describe the methodology used to search for and select the literature. This undermines the transparency and reproducibility of the analysis. The methods section should be supplemented with a brief description of the search strategy (databases, years, keywords).

RESPONSE: Our review was not sistematic. We didn’t have a Search strategy

  1. The authors mention the controversy surrounding the effectiveness of HA, but there is no detailed analysis of the differences in study results (e.g., diverse patient populations, OA severity criteria, different administration regimens). The discussion section should be expanded to discuss the sources of heterogeneity and their impact on the interpretation of the results.

RESPONSE: We introduced the following paragraph: The controversy surrounding the clinical effectiveness of viscosupplementation is largely fueled by the heterogeneity of the available evidence. Results from clinical trials and subsequent meta-analyses have been inconsistent, leading to divergent recommendations in clinical practice guidelines[10, 25, 26]. This variability can be attributed to several factors, including the pooling of data from studies with diverse patient populations, different criteria for osteoarthritis severity (e.g., Kellgren-Lawrence grades I-IV), and varied administration regimens (e.g., single vs. multiple injections). Crucially, many large-scale analyses fail to differentiate between the distinct biochemical properties of the various HA products. This methodological issue has contributed to conclusions from highly influential meta-analyses, such as the 2022 review by Pereira et al. and the analysis informing the American Academy of Orthopaedic Surgeons (AAOS) guidelines, which found that the average treatment effect of HA did not meet the threshold for clinical significance, thus recommending against its routine use[9, 26]. (lines 197-210)

  1. The paper does not address the economic issues that are important in the choice of OA therapy. Please at least briefly refer to the available pharmacoeconomic analyses or indicate the need to conduct them.

RESPONSE: We indicated the need to conduct pharmacoeconomic analyses (line 378)

  1. Although the article discusses biological and clinical aspects, there is a lack of information about the availability of various HA preparations on the market, registration requirements, or differences between countries. Please supplement the work with a practical element that will facilitate the application of the conclusion in everyday medical practice.

RESPONSE: We didn't want to get into the commercial field. We also found it very difficult to talk about the different realities in different countries.

  1. In section 4, there are two subsections marked as 4.1. Please correct the numbering to maintain a logical and consistent structure of the paper.

There are typos and minor linguistic errors in the text (e.g., “Futhermore” → “Furthermore,” “foreing” → “foreign,” ‘Classicaly’ → “Classically,” “It is been shown” → “It has been shown”). Please proofread the entire manuscript.

Repetitive and overly categorical phrases (“the conclusion is evident,” “most advanced and logically superior”) are noticeable. I suggest softening the tone towards a more balanced scientific style.

Please standardize the spelling: “intra-articular” vs. “intraarticular.”

Abbreviations and definitions should be used consistently – for example, UHMW (Ultra-High Molecular Weight) should be defined uniformly and used throughout the text.

Table 1 requires correct formatting (header, column layout, consistent use of units).

For greater clarity, I recommend adding diagrams or figures (e.g., comparison of linear and cross-linked preparations, HA mechanism of action).

There are inconsistencies in the formatting of the reference list: different abbreviations for journals, no spaces after commas, inconsistent presentation of DOI numbers (some works include DOI, others do not). Please standardize in accordance with the journal's style.

In the “Conflicts of Interest” section, the phrase “The author declares...” appears, while the article has two authors. Please correct it to “The authors declare...”.

Please standardize the notation of units and numerical values – in some parts of the text, “MDa” is used, while in other places the verbal notation (“Million Dalton”) is used. It is recommended to use a uniform abbreviated form.

Please ensure that all abbreviations are expanded when first used in the text and used consistently.

RESPONSE: Done, thank you!

Reviewer 3 Report

Comments and Suggestions for Authors

In this review paper focused on viscosupplementation for osteoarthritis, the authors criticize a system of categorizing hyaluronic acid (HA) products by molecular weight that has long been outdated. They introduce a new two-tiered classification system that first distinguishes between linear (non-modified) HA and cross-linked (chemically modified) HA, with linear HA being categorized depending on molecular weight. The authors emphasize the importance of bioactivity over purely mechanical activity of HA, and describe how HA can have benefits for the inflammatory signaling pathway, chondro-protective neurological signal, and pain signal via receptor signaling, regardless of the molecular structure. The authors maintain that high molecular weight linear HA has the highest safety profile, greatest biocompatibility, and the longest effect, relative to cross-linked HA, and can be used as a scaffold for corticosteroids, PRP, and orthobiologics. They consider this form of HA to be the 'gold standard' for the management of osteoarthritis. However, authors should address the following comments.

1- Originality of Proposed Classification: The authors offer a unique two-tier classification system by differentiating between linear and cross-linked HA and, for the linear HA, they further divide it by molecular weight. This is presented as a novel classification system, however, the authors' two-tier approach resembles some previously proposed classification systems in the literature by distinguishing between native vs modified HA based on chemical structure. The authors should compare their proposed classification with previous classification systems (e.g., Altman et al. (2015) or Peck et al. (2021)) and demonstrate that it is indeed a superior classification at reducing confusion, rather than a claim without evidence of any improved clinical utility.

2- The review suggests a broad change from patterns of mechanical action to patterns of biological action in the action of HA, citing interactions with receptors like CD44 or RHAMM. Although these actions garner some support by references, it overlooks measured rheological data because, for example, there are no studies or comparisons of the viscoelastic properties (e.g., zero-shear viscosity) of high-MW linear HA vs HA cross-linked with various agents, using measures such as dynamic moduli, from rheometry studies. The absence of those rheological measures undermines the argument in this article because mechanical effects, both healthy vs diseased joints, could potentially still be dominant in the short-term relief seeing that technical studies (e.g. oscillatory shear testing) should be included to draw a complete narrative.

3- There is no strong rationale for classifying linear HA into low (<1MDa), intermediate (1-2 MDa), and high (>2 MDa) MW. Why > 2 MDa specifically for HMW when some studies suggest that there is optimal biological activity with MW around 0.5-3 MDa (e.g., Maheu et al., 2016)? This cut-off seems arbitrary and disputable based on the non-linear relationship between MW and binding affinity for CD44, as highlighted by the existing in vitro dosage response curves from receptor-ligand binding assays.

4- While the claim that high concentration, high molecular weight linear HA provides effect for one year (based on the SOYA trial [23]) is persuasive, it is based on one open-label study, which may include bias (e.g., no blinding). It is suggested that head-to-head RCTs with cross-linked HA should be used, as there may olik in the pharmacokinetics (e.g., intra-articular half life by radiolabelled) not support the conclusion of equivalent residence time without chemical modifications (Rezende and Campos, 2012, etc.).

5- The review addresses order of increased risk of pseudosepsis and foreign-body reactions from cross-linked HA (e.g. Hylan G-F 20) with some case reports used to support this assertion. It is possible that this overstates the situation as meta-analyses (for example, Bannuru et al., 2019) demonstrate that adverse event rates for linear an cross-linked formulations were not statistically different when accounting for dose. A quantitative risk-benefit analysis based on odds ratios from pooled data would represent a more constructive technical critique, rather than somewhat taking the review's work out of context by taking an inordinate focus on reported infrequent events.

6- Evidence suggesting that linear HA, particularly around 0.5-3 MDa, produces optimal chondroprotection by decreasing the expression of ADAMTS-5 and MMP is noted from in vitro work in the literature (e.g. Zhou et al 2008). This largely ignores that the high-MW HA will have fragmented before and/or after injection, to the extent that low-MW pieces may lead to inflammation (Lee et al 2021). Differences in results across large animal models (e.g. rabbits with ACL transection) and gel electrophoresis flow cytometry methods to analyze fragments from swabs, show that reliably demonstrating long-term chondoprotective effects from clinical sterilized HA injections, without pro-catabolic effects, still requires more investigation.

7- While the analgesic effects attributed to TRPV1/ASIC masking and substance P decrement are presented, the authors neglect to disclose HA's binding constants (Kd values) with these channels. This is a potential technical limitation, as low-affinity interactions may not be sufficient in acidic environments ordinarily found in OA (pH ~6.5). To challenge the contribution of direct compared to indirect (through reduction of inflammation) effects, the authors could make reference to electrochemical or patch-clamp studies.

8- The review center around knee OA, possibily because the AAOS and OARSI guidelines are specific to the knee as a joint. Claims of HMW linear HA as a "benchmark" in regards to OAd, evidence is limited since multicenter trails or and biomechanical simulations referring to joint specific loading such as finite element analysis of shear stress, focused on the knee joint is required prevent overgeneralization.

9- References are too old and should update with new releavant references. 

Author Response

Dear reviewer,

Thank you so much for your comments.

Here is a point-by-point response to them:

  • Originality of Proposed Classification: The authors offer a unique two-tier classification system by differentiating between linear and cross-linked HA and, for the linear HA, they further divide it by molecular weight. This is presented as a novel classification system, however, the authors' two-tier approach resembles some previously proposed classification systems in the literature by distinguishing between native vs modified HA based on chemical structure. The authors should compare their proposed classification with previous classification systems (e.g., Altman et al. (2015) or Peck et al. (2021)) and demonstrate that it is indeed a superior classification at reducing confusion, rather than a claim without evidence of any improved clinical utility.

RESPONSE: . “We will then challenge the traditional division of viscosupplementation products primarily into native vs. cross-linked and low-molecular-weight vs. high-molecular-weight formulations[14] and propose a new, more clinically and biologically relevant framework.”

We acknowledge the previous systems and explained further why we think ours is more relevant

  • The review suggests a broad change from patterns of mechanical action to patterns of biological action in the action of HA, citing interactions with receptors like CD44 or RHAMM. Although these actions garner some support by references, it overlooks measured rheological data because, for example, there are no studies or comparisons of the viscoelastic properties (e.g., zero-shear viscosity) of high-MW linear HA vs HA cross-linked with various agents, using measures such as dynamic moduli, from rheometry studies. The absence of those rheological measures undermines the argument in this article because mechanical effects, both healthy vs diseased joints, could potentially still be dominant in the short-term relief seeing that technical studies (e.g. oscillatory shear testing) should be included to draw a complete narrative.

RESPONSE: We added some new references to further back up this section

  • There is no strong rationale for classifying linear HA into low (<1MDa), intermediate (1-2 MDa), and high (>2 MDa) MW. Why > 2 MDa specifically for HMW when some studies suggest that there is optimal biological activity with MW around 0.5-3 MDa (e.g., Maheu et al., 2016)? This cut-off seems arbitrary and disputable based on the non-linear relationship between MW and binding affinity for CD44, as highlighted by the existing in vitro dosage response curves from receptor-ligand binding assays.

We believe the rationale for the linear LMW classification is very strong, since “low-molecular-weight linear products may be considered suboptimal, as they are characterized by reduced efficacy and a proinflammatory nature.”

In general articles consider products below 500-730kD as LMW.

But yes, our cut-off is arbitrary. It based part in the literature, part in our feeling and experience in the field.

 “Conversely, both high-molecular-weight linear and cross-linked products exhibit a more favorable durability and efficacy profile”. We wanted to clearly separate the products, leaving the intermediate as a “grey zone”

  • While the claim that high concentration, high molecular weight linear HA provides effect for one year (based on the SOYA trial [23]) is persuasive, it is based on one open-label study, which may include bias (e.g., no blinding). It is suggested that head-to-head RCTs with cross-linked HA should be used, as there may olik in the pharmacokinetics (e.g., intra-articular half life by radiolabelled) not support the conclusion of equivalent residence time without chemical modifications (Rezende and Campos, 2012, etc.).

RESPONSE: We acknowledge in lines that Future research should focus on head-to-head trials comparing HMW Linear HA against Cross-linked HÁ

  • The review addresses order of increased risk of pseudosepsis and foreign-body reactions from cross-linked HA (e.g. Hylan G-F 20) with some case reports used to support this assertion. It is possible that this overstates the situation as meta-analyses (for example, Bannuru et al., 2019) demonstrate that adverse event rates for linear an cross-linked formulations were not statistically different when accounting for dose. A quantitative risk-benefit analysis based on odds ratios from pooled data would represent a more constructive technical critique, rather than somewhat taking the review's work out of context by taking an inordinate focus on reported infrequent events.

We don’t agree that serious adverse effects are infrequent with cross-linked products. There are several case reports. Not even one with linear products.

6- Evidence suggesting that linear HA, particularly around 0.5-3 MDa, produces optimal chondroprotection by decreasing the expression of ADAMTS-5 and MMP is noted from in vitro work in the literature (e.g. Zhou et al 2008). This largely ignores that the high-MW HA will have fragmented before and/or after injection, to the extent that low-MW pieces may lead to inflammation (Lee et al 2021). Differences in results across large animal models (e.g. rabbits with ACL transection) and gel electrophoresis flow cytometry methods to analyze fragments from swabs, show that reliably demonstrating long-term chondoprotective effects from clinical sterilized HA injections, without pro-catabolic effects, still requires more investigation.

  • While the analgesic effects attributed to TRPV1/ASIC masking and substance P decrement are presented, the authors neglect to disclose HA's binding constants (Kd values) with these channels. This is a potential technical limitation, as low-affinity interactions may not be sufficient in acidic environments ordinarily found in OA (pH ~6.5). To challenge the contribution of direct compared to indirect (through reduction of inflammation) effects, the authors could make reference to electrochemical or patch-clamp studies.

RESPONSE: We honestly haven't done such an in-depth review to discuss this type of information.

  • The review center around knee OA, possibily because the AAOS and OARSI guidelines are specific to the knee as a joint. Claims of HMW linear HA as a "benchmark" in regards to OAd, evidence is limited since multicenter trails or and biomechanical simulations referring to joint specific loading such as finite element analysis of shear stress, focused on the knee joint is required prevent overgeneralization.

We agree that we went to far claiming HMW as a “benchmark”and rewrote the article with more balanced arguments

  • References are too old and should update with new releavant references. 

RESPONSE: done

Round 2

Reviewer 1 Report

Comments and Suggestions for Authors The authors have addressed my previous comments satisfactorily and implemented the requested revisions.  I have no further substantive concerns.

Reviewer 3 Report

Comments and Suggestions for Authors

the article has well revised and suitable for publication now.